# Anti-Inflammatory Properties of *Bellevalia saviczii* Root Extract and Its Isolated Homoisoflavonoid (*Dracol*) Are Mediated by Modification on Calcium Signaling

**DOI:** 10.3390/molecules24183376

**Published:** 2019-09-17

**Authors:** Monica Savio, Mohammed Farhad Ibrahim, Chiara Scarlata, Matteo Orgiu, Giuseppe Accardo, Abdullah Shakur Sardar, Francesco Moccia, Lucia Anna Stivala, Gloria Brusotti

**Affiliations:** 1Department of Molecular Medicine, Immunology and General Pathology Unit, via Ferrata 9, University of Pavia, 27100 Pavia, Italy; monica.savio@unipv.it (M.S.); chiara.scarlata01@universitadipavia.it (C.S.); giuseppe.accardo01@universitadipavia.it (G.A.); 2Department of Drug Sciences, viale Taramelli 12, University of Pavia, 27100 Pavia, Italy; gardy1988@gmail.com (M.F.I.); gloria.brusotti@unipv.it (G.B.); 3Department of Environmental Science, College of Science, University of Salahaddin-Erbil, Erbil 44001, Iraq; 4Department of Biology and Biotechnology “L. Spallanzani” via Forlanini 6, University of Pavia, 27100 Pavia, Italy; matteo.orgiu01@universitadipavia.it (M.O.); francesco.moccia@unipv.it (F.M.); 5Department of Biology, College of Education, University of Salahaddin-Erbil, Erbil 44001, Iraq; abdullah.sardar@su.edu.krd

**Keywords:** *Bellevalia saviczii*, homoisoflavoinoids, NF-kB, Ca^2+^ signaling

## Abstract

*Bellevalia saviczii* is a medicinal plant used as anti-rheumatic and anti-inflammatory herbal remedy in Iraqi-Kurdistan. The aim of this study was to evaluate the anti-inflammatory activity of its extract and the isolated homoisoflavonoid (*Dracol*) by studying the Ca^2+^-dependent NF-kB pathway. Nuclear translocation of p65 NF-kB subunit, as parameter of NF-kB activation, was visualized in human leukemic monocytes by immunofluorescence and Western blot analyses, after cell treatment with *B. saviczii* root extract or *Dracol* followed by Lipopolysaccharide stimulation. In parallel, Ca^2+^ signals responsible for NF-kB activation and levels of inflammatory cytokines were investigated. LPS-induced p65 translocation was evident in monocytes and both treatments, in particular that with *Dracol*, were able to counteract this activation. Intracellular Ca^2+^ oscillations were halted and the cytokine release reduced. These results confirm the traditional anti-inflammatory efficacy of *B. saviczii* and identify one of the molecules in the extract which appears to be responsible of this action.

## 1. Introduction

*Bellevalia saviczii*, a perennial plant belonging to the family of *Asparagaceae*, represents a plant genus particularly rich in homoisoflavonoids, a special subclass of flavonoids, rarely found in nature but responsible for the biological activity of *Bellevalia* species used in traditional medicine by people from Mediterranean to Central Asia [1,2]. In particular, *B. saviczii* is one of the most common local plants of Kurdistan. Traditionally, as handed down orally, it is application as wraps and infusions anti-rheumatic and anti-inflammatory remedy, however their use has not been documented yet in any reports. Homoisoflavonoids, abundant in genus *Bellevalia*, have received much attention over the past 30 years in the research and development of natural bioactive compounds. Until now, about 240 homoisoflavonoids have been isolated and identified, which could be classified into five structural categories, sappanin, scillascillin, brazilin, caesalpin and protosappanin-types. They have been reported with a broad range of bioactivities, including anti-microbial, anti-mutagenic, immunomodulatory, anti-diabetic, cytotoxic [3,4], antioxidant [5,6,7], anti-angiogenic, vasorelaxant and anti-inflammatory effects [8,9,10]. In the sappanin-type, 2,3-dihydro-3,5-dihydroxy-7-methoxy-3-[(4-methoxyphenyl)-methyl]-8methyl-4*H*-[1]benzopyr-an-4-one, for which the trivial name *Dracol* has been proposed, is a natural compound isolated for the first time by Hernandez et al. in 2006 from genus *Dracaena draco* together with another compound namely icodeside [3]. Whereas this last compound showed moderate cytotoxicity against human HL-60 and A-431 cells, as determined by the MTT assay after treatment with different concentrations for 72 h, *Dracol* was completely ineffective (IC_50_ > 100 µM).

Considering that the most common use in traditional medicine of *B. saviczii* extract is as anti-inflammatory remedy, in this work its potential anti-inflammatory properties were investigated by studying the nuclear factor-kB (NF-kB) activation in human leukemic monocytic cell line (THP1). This protein is included in a family of transcription factors implicated in inflammation, immune response, cell survival and cancer [1,2]. At the basal level, NF-kB is localized in the cytoplasm and its activity is normally suppressed by the interaction with IkB inhibitory proteins, which thereby mask NF-kB nuclear localization signals [11,12,13]. However, in response to specific external stimuli, including pro-inflammatory cytokines like TNFα, IL1β or endotoxins (Lipopolysaccharide, LPS), viral infection, oxidants, phorbol esters and ultraviolet irradiation, the IkB component of the complex is phosphorylated by IKKβ, subsequent degraded and ubiquitinated, resulting in translocation of NF-kB into the nucleus and the induction of target gene transcription [14]. This nuclear translocation of the p50-p65 subunits of NF-kB triggers pro-inflammatory cytokine gene expression such as inducible nitric oxide synthase (iNOS), cyclooxygenase 2 (COX2), TNFα, IL1β and IL6. The nuclear translocation of NF-kB may also be triggered by oscillations in intracellular Ca^2+^ concentration ([Ca^2+^]_i_), which recruit the Ca^2+^-dependent decoders calcineurin and calmodulin to promote IkB degradation [15,16,17,18]. Notably, LPS was found to recruit NF-kB through an oscillatory Ca^2+^ signal in several cell types, such as primary rat lung microvascular endothelial cells [19] and mouse microglial cells [20]. Intracellular Ca^2+^ oscillations provide the most suitable waveform to selectively engage NF-kB rather than other Ca^2+^-dependent transcription factors, such as the nuclear factor of activated T cells (NFAT), and permit persistent nuclear NF-kB expression [21]. The role of Ca^2+^ signaling in LPS-induced NF-kB activation in THP1 cells, as well as in human primary monocytes, remains, however, unclear.

Aberrant NF-kB activity is associated with various inflammatory diseases including arthritis, cancer, and atherosclerosis [6,7,10,11,12]. Thus, in the need of a more effective therapy for the treatment of inflammatory diseases, specific inhibition of p65 translocation, related to reduced NF-kB activity represents a rational target [14]. 

Therefore, this study aims at investigating the anti-inflammatory properties of the *B. saviczii* root methanol extract (BRME) and its isolated compound (*Dracol*) against LPS-induced inflammatory *stimulus* in THP1 cells. Here, we report that both extract and its component *Dracol* protected against inflammatory response induced by LPS. These findings, for the first time, demonstrate that using *B. saviczii* plant extract, containing bioactive compounds (homoisoflavonoids), might be beneficial to counteract inflammatory diseases. 

## 2. Results

The aim of this investigation was to establish whether BRME and its isolated compound, *Dracol*, are able to contrast LPS-induced inflammatory response of leukemic monocytes, thereby confirming the documented anti-inflammatory properties of this plant in traditional medicine. The extraction of *B. saviczii* root, the purification of the homoisoflavonoids *Dracol* and its characterization by HPLC-DAD MS method has been described in Figure 1 and Appendix A. 

The proposed experimental model, consisting of the nuclear translocation of p65 NF-kB subunit, has been already used [13] and represents a simple and useful tool to study anti-inflammatory activity of bioactive compounds. The evaluation of p65 NF-kB subunit, both by immunofluorescence and Western blot analysis, was coupled to functional evaluation of THP1 cell viability and proliferation. In addition, the levels of inflammatory cytokines and calcium signaling were investigated trying to clarify the mechanism of action of *B. saviczii* root extract and its isolated compound *Dracol*. 

### 2.1. Cell Viability and Proliferation Assays

Preliminary experiments, performed to study the cytotoxic effect of BRME and *Dracol* by the MTT test (Figure 2A), showed that both do not significantly affect THP1 cell survival or growth at all the concentrations used, after 24 h of treatment. These results were in agreement with those of Hernández et al. [3] that demonstrated a completely inefficacy of *Dracol* in exerting toxicity on HL-60 and A-431 cells. Due to this non-toxic effect (about 10%), we chose the concentrations of 10 and 100 μM for *Dracol* and 62.5 up to 500 µg/mL for BRME for all the subsequent experiments. 

The MTT is a test for assessing not only cell toxicity and death but also cell proliferation. However, to better distinguish cytotoxic from cytostatic effect of the compounds, DNA synthesis was evaluated by BrdU incorporation and flow cytometry analysis. The treatment of 24 h with BRME and *Dracol* did not result in any significant change in THP1 cell proliferation (Figure 2B,C).

These data are in contrast with Valente et al. that demonstrated antiproliferative activity of *D. draco* leaf and fruit extracts (0–400 µg/mL) in human colon and renal tumour cells in vitro, whereas a weak effect was observed in HepG2 cells. The highest activity was observed with the leaf extract [22]. Trypan blue staining confirmed this weak cytotoxic effect that was related to low level of cell death, even at the higher concentrations of 200 μM *Dracol* and 1000 μg/mL BRME, and a longer incubation time (48, 72 and 96 h) (Appendix A).

### 2.2. Optimization of Inflammatory Model: Preliminary Evaluation

Previous experiments on THP1 cells treated with different concentrations allowed to find the best LPS concentration to induce NF-kB activation. In particular, THP1 cells were treated in serum free medium with 0.01–0.1–1 µg/mL LPS for 4 h. As shown in Figure 3A,B, a dose-dependent increase of NF-kB positive nuclei was observed; in these conditions, the viability remains rather high, showing a 20% of reduction only at the highest LPS concentration. No differences were evident at both incubation time used, 2 and 4 h (Figure 3C). 

Only a weak cytotoxic effect related to a low number of dead cells (about 20%) was detectable at the highest concentration of LPS (Figure 3B). Therefore, 4 h 1 µg/mL LPS treatment was considered in the subsequent experiments. The p65 translocation using immunofluorescence analysis has been used by Hund et al. [13], but with different cell type and inflammatory *stimulus*; in particular, they used endothelial cells and TNFα, respectively. 

### 2.3. Effect of BRME and Dracol on LPS-Activated THP1

Before determining the effects of BRME and *Dracol* on LPS-induced cytotoxicity and NF-kB activation, THP1 cells were treated with each compound alone for 24 h to check for a possible effect on NF-kB. The results showed that they do not exert any modulation on NF-kB activity (Appendix A). One µg/mL LPS for 4 h confirmed a weak not significant reduction of viability (about 10%) as shown in Figure 4A. 

While BRME was able to counteract this cytotoxicity, *Dracol* (10–100 µM) seems to have no effect as compared with only LPS treated cells. Both *Dracol* and BRME at all concentrations used, were able to reduce in a highly significant manner the LPS-induced NF-kB activation (Figure 4B,C), as demonstrated both by immunofluorescence staining and Western blotting experiments (Figure 4D). These data, for the first time, support the traditional anti-inflammatory properties of *B. saviczii*, showing that both total root extract and the isolated compound *Dracol* prevent THP1 inflammatory response. Inflammation has been considered as a concourse of several diseases, such as type 2-diabetes, cardiovascular, neurodegenerative and neoplastic diseases [23,24,25]; therefore, it is important to highlight potent inflammatory modulators present in herbal remedies. 

### 2.4. Cytokine Analysis

To better characterize the anti-inflammatory properties of *Dracol* and BRME, the levels of TNFα, IL6 and IL1β were also evaluated. In agreement with the data obtained for NF-kB activation, the level of TNFα was significantly increased after treatment with LPS, as shown in Figure 5A; both *Dracol* and BRME, at all concentrations tested, significantly inhibited the release of the pro-inflammatory cytokine. While, similar results were obtained for IL6 and IL1β (Figure 5B,C), in which only the highest concentrations of both *Dracol* and BRME are able to significantly inhibit cytokine release. 

### 2.5. Calcium Signalling 

It has long been known that intracellular Ca^2+^ oscillations drive the nuclear translocation of NF-kB by favouring the Ca^2+^-dependent phosphorylation of IKB [15,16,17,18]. LPS, in turn, is able to induce repetitive intracellular Ca^2+^ oscillations by inducing a concerted interplay between inositol-1,4,-5-trisphosphate (InsP_3_)-dependent Ca^2+^ release and store-operated Ca^2+^ entry (SOCE), thereby inducing NF-kB activation [19,20,25]. We found that the acute addition of 1 µg/mL LPS caused an increase in [Ca^2+^]_i_ in the 54.1 ± 13.8 % (*n* = 377) of THP1 cells loaded with the Ca^2+^-sensitive fluorochrome, Fura-2/AM. This Ca^2+^ signal consisted either in a single Ca^2+^ spike (59.5 ± 17.7%) (Figure 6A) or in repetitive Ca^2+^ oscillations that persisted throughout the period of recording (40.5 ± 17.7%) (Figure 6B). 

In agreement with these observations, intracellular oscillations in [Ca^2+^]_i_ were detected in the majority of THP1 cells pre-treated with 1 µg/mL LPS for 4 h (Figure 6C,D), whereas the remaining cells displayed only one Ca^2+^ transient during the period of recording (not shown). These repetitive Ca^2+^ spikes were similar to those recorded in mouse pulmonary artery endothelial cells previously exposed to 1 µg/mL LPS for 16 h [26]. Of note, 1 µg/mL LPS failed to induce intracellular Ca^2+^ oscillations in THP1 cells pre-treated with BAPTA-AM (30 μM, 2 h) (Appendix A), which confirms that intracellular Ca^2+^ signaling was required to induce the nuclear translocation of NF-kB [27]. However, *Dracol* (10 μM) and BRME (250 μg/mL) reduced the percentage of oscillating cells (Figure 6C), although the statistical significance was achieved only upon pre-treatment with *Dracol* (Figure 6C). In addition, *Dracol* (10 μM) and BRME (250 μg/mL) decreased the frequency and amplitude of the 1^st^ Ca^2+^ spike of LPS-induced intracellular Ca^2+^ transients (Figure 6E,F). In the majority of non-excitable cells, intracellular Ca^2+^ oscillations are supported by rhythmical InsP_3_-dependent Ca^2+^ release from the endoplasmic reticulum (ER), the largest endogenous Ca^2+^ store, whereas SOCE is required to sustain the Ca^2+^ spikes over time by refilling the ER with the Ca^2+^ necessary to set up the next Ca^2+^ spike [19,20,26]. 

As *Dracol* and BRME affect the frequency and magnitude of LPS-induced Ca^2+^ spikes, further work will examine whether and how these compounds impair the expression and/or function of the components of the Ca^2+^ toolkit involved in the spiking response to LPS. 

## 3. Materials and Methods 

### 3.1. Reagents

LPS and all other chemicals of reagents grade were obtained from Sigma (St. Louis, MO, USA) unless otherwise specified. 

### 3.2. Plant Materials

*Bellevalia saviczii* was collected on April 2014 in the Zraraty district in Erbil-Kurdistan region. The materials were identified and classified at the Education Salahaddin University Herbarium (ESUH) by prof. Abdullah Shakur Sardar, of the University of Salahaddin, Erbil-Iraq. A voucher specimen (7702) was deposited. Roots were cleaned and air-dried in the shade at room temperature (r.t.) (20–25 °C). After drying, the plant material was grounded using a laboratory grinding mill and the resulting powder was stored in dark bottles at r.t. until required. 

### 3.3. Root Extraction and Compound Isolation

*B. saviczii* roots (100 g) were soaked in petroleum ether (500 mL) in an ultra-sonic bath for 30 min, and then left in the same solvent for 3 h under continuous stirring at r.t. The procedure was repeated three times. Defatted roots were subsequently soaked in methanol (MeOH) (500 mL) in an ultra-sonic bath for 30 min, then left in the same solvent for 3 h under continuous stirring at r. t. The procedure was repeated three times. The mixtures were then filtered and the solvent removed *under vacuum* in a rotary evaporator to afford (2.0 g) of crude MeOH residues extract (BRME) from roots.

Subsequently, BRME fraction (2.0 g) was suspended in 500 mL of 40% aqueous MeOH and then extracted with dichloromethane (500 mL) to afford a dichloromethane soluble fraction (459.6 mg) which was called BRD. 459.6 mg of BRD were suspended in 10 mL of H_2_O/MeOH 80/20, and then charged on the C_18_ column. The first solvent system (H_2_O/MeOH 80/20) was added to the column and the elution started with gradient system to 100% MeOH at the flow rate of 10 mL/min; 500 mL were necessary to complete the elution of fractions. The column was finally washed with 165 mL of 100% MeOH. After evaporation of the solvent *under vacuum*, BRD1 (121.4 mg), BRD2 (217.3 mg), BRD3 (76.9 mg), BRD4 (9.17 mg) and BRD5 (12.8 mg) fractions were obtained.

The fraction BRD4 (9.17 mg), eluted (H_2_O/MeOH 3/7) on a Merck Aluminum-backed RP-18 TLC plate and revealed as a yellow spot upon exposure to the vanillin-sulphuric acid reagent and gentle heating, resulted to be a pure compound, later identified as compound **(1)** named *Dracol* (Scheme 1).

### 3.4. Spectral Data of Isolated Compound

Compound (**1**): ( = (3R)-2,3-Dihydro-3,5-dihydroxy-7-methoxy-3-[(4-methoxyphenyl) methyl]-8-methyl-4H-[1]-benzopyran-4-one). Pale yellow amorphous solid, yield: (9.17 mg), Molecular formula C_19_H_20_O_6_ deduced from the negative-ion ESIMS^-^ spectrum ([M − H]^−^
*m*/*z* 345.28, 239.25); UV (MeOH) λ_max_ 290.5, 375.5 nm; ^1^H-NMR (CD_3_OD, 300 MHz) 3.99 (Hβ, d, *J* = 12.0, H-2), 4.04 (Hα, d, *J* = 12.0, H-2), 6.15 (1H, s., H-6), 2.92 (2H, d, *J* = 5.5, H-9), 1.95 (3H, 2, H-8 Me), 3.76 (3H, s, OMe), 3.87 (3H, s, OMe), 6.82 (1H, d, *J* = 8.6, H-2′), 7.14 (1H, d, *J* = 8.6, H-3′), 7.14 (1H, d, *J* = 8.6, H-5′), and 6.82 (1H, d, *J* = 8.6, H-6′).^13^C-NMR (CD_3_OD, 300 MHz): δ_C_ 14.51 (C-8 Me), 61.7 (OMe), 61.9 (OMe), 70.9 (C-2), 62.9 (C-3), 201.5 (C-4), 146.9 (C-5), 105.16 (C-6), 149.7 (C-7), 130.3 (C-8), 33.1 (C-9), 135.5 (C-4a), 151.72 (C-8a), 121.8 (C-1′), 131.9 (C-2′), 116.7 (C-3′), 157.6 (C-4′), 116.7 (C-5′) and 131.5 (C-6′). The ^1^H and ^13^C-NMR spectra were in agreement with the literature [3].

### 3.5. In Vitro Cell Culture and Treatments 

Human leukemic monocytic cell line (THP1) provided by Istituto Zooprofilattico di Brescia (Italy) were cultured in RPM1 1640 medium with 10% of FBS at 37 °C in a humidified CO_2_ (5%) incubator.

Fifty-four mg of BRME were dissolved in Dimethyl sulfoxide (DMSO) at the concentration of 720 µg/µL. *Dracol*, isolated from BRME, was prepared as a stock solution in DMSO (150 mM) and diluted directly in cell culture medium. Final concentration of DMSO did not exceed 0.15% (*v*/*v*) and control cells were treated with the same concentration of vehicle that did not exert any effect in all the assays. 

THP1 cells have been treated for 24 h with different concentrations of the total root extract or *Dracol*. To perform the immobilization of cell suspension cultures, at the end of each treatment, the cells were washed with PBS and resuspended in serum-free medium before transferring to a culture plate containing coverslips [28]. LPS was used to set up the cellular inflammatory model [6,29] and preliminary experiments were performed to optimize timing and dosages of LPS treatment for activating THP1 cells. In particular, three increasing concentrations of LPS (0.01–0.1–1 µg/mL) and two different incubation times were used to identify the better condition for the subsequent experiments. The negative controls consisted of untreated cells while the positive ones were LPS-treated. From these preliminary approaches, THP1 cells treated for 24 h with different concentrations of BRME or *Dracol* and then stimulated with LPS (1 µg/mL for 4 h) provided an efficient and reproducible model to study monocyte inflammatory response. In addition, THP1 cells were treated for 2 h with the calcium chelator BAPTA-AM (30 µM) alone and/or in combination with LPS (1 µg/mL) in the above conditions. 

### 3.6. Cytotoxicity, Proliferation and Trypan Blue Exclusion Assay

The cytotoxicity of BRME and *Dracol*, both by themselves and in combination with LPS, was evaluated by MTT assay and Trypan Blue staining, as previously described [30]. Cell cycle experiments were performed by using 5-bromo-2′-deoxyuridine incorporation (BrdU), as previously reported [31].

### 3.7. NF-kB Staining 

As a marker of NF-kB activation, the nuclear translocation of the p65 NF-kB subunit was visualized in THP1 cells by immunofluorescence microscopy and Western blotting. After 4 h treatment with LPS in serum free medium, cells seeded on coverslips and pre-incubated for 24 h with compounds, were fixed in 4% PFA and 70% ethanol and then stored overnight at −20 °C. After washing with PBS, the samples were blocked in PBS containing 5% FBS, and then incubated overnight at 4 °C in a humidified chamber with the specific monoclonal anti-NF-kB p65 antibody (F-6) (sc-8008 Santa Cruz) diluted 1:200 in PBS. After washing, each reaction was followed by incubation for 1 h with anti-mouse conjugated with Alexa Fluor 488 (1:200; Invitrogen Molecular Probes, USA). The cells were incubated with Hoechst 33258 (1:5000) for 5 min at r.t. and then washed with PBS. Slides were mounted in Mowiol (Calbiochem) containing 0.25% 1,4-diazabicyclo-[2,2,2]-octane (Aldrich) as antifading agent. Images of the fixed THP1 cells were taken with Olympus IX 83(60/1.24) or OlympusBX-51 microscopes (100X immersion oil lens, NA 1.25) equipped with an Olympus C4040 camera. The concentration of NF-kB was also determined by Western blotting using the p65 antibody (F-6) (sc-8008 Santa Cruz) diluted 1:1000 in PBS/Tween20 (0.2%), as previously described [30].

### 3.8. Cytokine Analysis

At the end of the cell treatment, as above described, the supernatant of the centrifuged media was collected and stored at −80°C. Concentrations of TNFα and IL6 were determined by enzyme-linked immunosorbent assay (ELISA) according to the manufacturer’s protocol (Thermo Scientific, USA), using the antibodies C-4, sc-133192 and E4, sc-28343 (Santa Cruz, Biotechnology Inc. Heidelberg, Germany), respectively. The concentration of IL1β was determined by western blotting using the antibody E7-2-hIL1β (sc-32294, Santa Cruz, Biotechnology Inc. Heidelberg, Germany), as previously described [30]. Densitometric analysis for IL1β was performed on both bands obtained, considering that 31 and 17 KDa are the precursor and mature forms of the protein, respectively. 

### 3.9. [Ca^2+^]_i_ Measurements

LPS-induced intracellular Ca^2+^ oscillations were measured as described elsewhere [18]. At the end of the treatment for 4 h with LPS (1 µg/mL), THP1 cells were loaded with 4 µM fura-2 acetoxymethyl ester (Fura-2/AM; 1 mM stock in DMSO in physiological salt solution (PSS) for 20 min at 37 °C, 5% CO_2_ saturated humidity. When the cells were pre-treated for 24 h with 10 µM *Dracol* and 250 µg/mL BRME, LPS (1 µg/mL) was added for 4 h at the end of the incubation period and the Ca^2+^ signals then evaluated. PSS had the following composition (in mM): 150 NaCl, 6 KCl, 1.5 CaCl_2_, 1 MgCl_2_, 10 Glucose, 10 Hepes and was titrated to pH 7.4 with NaOH. The osmolality of PSS as measured with an osmometer (Wescor 5500, Logan, UT) was 338 mmol/kg.

After washing in PSS, the coverslips were fixed to the bottom of a Petri dish and the cells observed by an upright epifluorescence Axiolab microscope (Carl Zeiss, Oberkochen, Germany), usually equipped with a Zeiss ×40 Achroplan objective (water-immersion, 2.0 mm working distance, 0.9 numerical aperture). The cells were excited alternately at 340 and 380 nm, and the emitted light was detected at 510 nm. A first neutral density filter (1 or 0.3 optical density) reduced the overall intensity of the excitation light and a second neutral density filter (optical density = 0.3) was coupled to the 380 nm filter to approach the intensity of the 340 nm light. A round diaphragm was used to increase the contrast. The excitation filters were mounted on a filter wheel (Lambda 10, Sutter Instrument, Novato, CA, USA). Custom software, working in the LINUX environment, was used to drive the camera (Extended-ISIS Camera, Photonic Science, Millham, UK) and the filter wheel, and to measure and plot on-line the fluorescence from 30–45 rectangular “regions of interest” (ROI) enclosing 20–30 single cells. Each ROI was identified by a number. Adjacent ROIs never superimposed. [Ca^2+^]_i_ was monitored by measuring, for each ROI, the ratio of the mean fluorescence emitted at 510 nm when exciting alternatively at 340 and 380 nm [Ratio (F_340_/F_380_)]. An increase in [Ca^2+^]_i_ causes an increase in the ratio [18]. Ratio measurements were performed and plotted on-line every 3 sec. The experiments were performed at r.t. (22 °C). 

### 3.10. Statistical Analysis

Data are presented as means ± SD. Statistical analysis was performed by using the Student’s *t*-test where only probability values * *p* ≤ 0.05, ** *p* ≤ 0.01 were considered to be statistically significant. Intracellular Ca^2+^ oscillations were analyzed by measuring the percentage of oscillating cells for each condition (LPS, LPS + BRME, LPS + *Dracol*), the amplitude of the 1^st^ spike and the frequency of Ca^2+^ over 1000 s. Pooled data are presented as means ± SE and the number of analyzed cells is reported above or within column bars. Each data is representative of four different coverslips. Statistical significance (*p* ≤ 0.05) was evaluated by the Student’s *t*-test for unpaired observations.

## 4. Conclusions

The present study demonstrated that the root of *B. saviczii*, which has been traditionally used in Iraqi-Kurdistan as a medicinal plant, is a valuable source of active compounds, especially homoisoflavonoids. Among them, *Dracol* has been identified for the first time in this plant and, in our experimental conditions, it is able to prevent the NF-kB activation induced in human monocytes by LPS treatment. Very few studies have considered the biological activity of this compounds, which deserves further investigation, particularly regarding their use in treating inflammatory diseases. The observed decrease in nuclear translocation of p65 NF-kB subunit, associated with TNFα release reduction, strongly depend on the modification of calcium signaling.

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
