# Peer review of "Anti-Inflammatory Properties of *Bellevalia saviczii* Root Extract and Its Isolated Homoisoflavonoid (*Dracol*) Are Mediated by Modification on Calcium Signaling"

_molecules, 2019, doi:10.3390/molecules24183376_

Round 1
Reviewer 1 Report
This study evaluated potential antiinflammatory effects of Bellevalia saviczii extract and its major component, dracol, in THP1 cells stimulated by LPS. Although the concept of this study is interesting, the quality of presented data is insufficient for publication in this journal.
Fig. 1. Please increase font size. It would be better if the separation process is explained in detail under the figure legend, rather than in the figure itself where space is limited. Then Figure can be made simpler using larger fonts.
Fig. 2. Please spell out all abbreviations in Figures where they appear for the first time. Statistical analysis results should be marked in all figures (Fig 2A and C). Font size is too small to read (especially in Fig 2B). Instead of showing all raw data, please show representative data in Fig 2B. Please explain the large difference between control and others in Fig 2B. Please include a positive control for cytotoxicity comparison (i.e., LPS of varied concentrations)
Fig. 3. In A, The image quality is not sufficient for use in fluorescence-based quantitative analysis of NF-kappaB activation. Cell density looks different between images. It is important to show cell images treated with test materials (extract or dracol) only, and in combination with LPS, but they were not shown.
Fig. 4. Inhibitory effects of extract or dracol on NF-kappaB activation are not proportional to the concentration of test materials (extract or dracol). How is it possible? It is important to show data from the cells treated with test materials (extract or dracol) in the absence of LPS, to know whether they alone stimulate NF-kB. A positive control is needed in all these experiments. Marks for statistical difference in Fig 4B are confusing. Please make them clear. Additional experiments using western blot are essentially needed to examine NF-kB modulation by LPS and test materials (extract or dracol).
Fig. 5. Inhibitory effects of extract or dracol on TNF-alpha level are not proportional to the concentration of test materials (extract or dracol). LPS increased IL6 level, and it was inhibited by 500 ug/ml of the extract, but it was enhanced by 125-250 ug/ml of the extract. How are these results possible? It is important to show data from the cells treated with test materials (extract or dracol) in the absence of LPS to know whether they alone affect TNF-alpha and IL6 levels. Marks for statistical difference in Fig 5A-B are confusing. Please make them clear.
Fig. 5C. The quality Western blot images are not so good to be used in quantitative analysis. The experiments should be repeated at least 3 times and statistical analysis of the results are essentially needed. LPS did not increase IL1beta. Is it true? Is this a special case or a general finding?
Fig. 6. Variation between experiments of the same treatment is very large. Please show calcium measurement data when cells were treated with calcium ionophore and BAPTA, as a positive or negative control. What are the numbers on the Fig. 6D-E? 243, 95, 57….? No statistical difference between treatments in Fig 6D? Marks for statistical difference in Fig 6E and F are confusing. Please make them clear. Is it true for (n=377) in the text? Test concentrations of the extract and dracol should be indicated.
Overall the data presented in the paper are insufficient to support the conclusion.
Reviewer 2 Report
General comments
The article deals with the experimental analysis of the effects of extracts from the root of Bellevalia saviczii plant on inflammation pathways affecting human leukemic monocytic cell, with focus on whole methanol extract and the homoisoflavonoid Dracol, the latter isolated from the specific raw material for the first time. The effect of the bioactive compounds on calcium signalling, altered by endotoxins (LPS), was identified as the primary anti-inflammatory mechanism.
The manuscript is sufficiently well structured, the experiments were correctly designed, the methods were correctly explained. The study is rigorous, novel, original, and informative.
Some issues remain for the structure of the manuscript, some phrasing, and the presentation of few of the results.
Specific comments
Line 22. "LPS". Define the abbreviation in the Abstract too.
Line 43. "[3-10]". Too many references together. As a support to the interested reader, try to distribute the references in association to the relevant effect among those listed above.
Lines 82-85 (up to "to Central Asia"). Remove or integrate in the final part of the Introduction.
Lines 85-89. This text, including Figure 1, concern the "Methods", thus I suggest moving to Section 3.
Line 102. "Since". Change to "Due to".
Lines 121-123 (up to "in HepG2 cells"). Any explanation about the reason for this contrast?
Lines 123-124. "(Valente et al., 2012). Format reference as per the journal's style.
Line 152 - Fig. 4. The abbreviation DMSO (Dimethyl sulfoxide, I guess) should be defined.
Line 166. "modulator". Change to "modulators".
Line 172. "Similar results were obtained for IL6". In Figure 5B, the pattern for IL6 looks like more complex than stated, for example with BRME, it increases from 62.5 to 125, then decreases, and only at the concentration of BRME at the level of 500 microg/mL it falls below the LPS sample... more explanation needed.
Line 198. "for 4 hours". Based on the scale shown in Figure 6A, the time covered looks like to be about one hour, anyway far less than 4 hours. Check and explain (also caption of Figure 6).
Line 209. "abolished". Too drastic and unsupported by the results. Change to "attenuated".
Line 262. "DMSO". This abbreviation (Dimethyl sulfoxide, I guess) should be defined at first appearance.
Line 338. "demonstrated that the B. saviczii". Change to "demonstrated that the root of B. saviczii".
Reviewer 3 Report
The paper reviewed bring novel knowledge to the understanding of the effects produced by traditional plant B. saviczii. Work described is of good quality, going in depth with insight into mechanism of action. One of my concerns is the distribution of a plant taken from Kurdistan. I would like to see here also if these effects are connected to the plant in general (collected from another geographical site or at different year).
Please note that on Figure 1 there is missing "using rotary ___" (evaporation is missing).
Did authors tried to add larger concentration of Dracol since "although 10 % reduction of cell viability was observed (Figure 2B-C)" and also found minor citotoxic effect using Trypan blue exclusion? And moreover why did you choose 24 h treatment and not longer periods? Repeated use of these compounds and herbal preparations may lead to extension of a period in which the cells are in contact with a plant.
Did author thought how to asses the possible uptake of a compound? Namely, you assessed effect of compounds in THP1 cell line. It would be more applicable if you did experiments in a cell line that is more "accessible" to the compound which are used topically! E.g. epithelial cell line.
Please find a usfull paper regarding Ca level: doi: 10.1007/s10534-014-9792-x.
Since you analyzed several groups in one experiment I find more accurate to use ANOVA with posthoc test than Student t-test.
Round 2
Reviewer 1 Report
The paper was improved much. I have some minor concerns.
Please replace the western blot image in Fig 5C, with the image from an "uncut" membrane. If you need sufficient time for experiments, you can contact editorial office. Please change the title of Y axis in Figure 5C. arbitrary unit (A.U) --> pro-IL1b (A. U.). Figure 5A, TNFa (%) --> TNFa (Fold) or TNF-a (A.U.) Figure 5B, IL-6 (%) --> IL-6 (Fold) or IL-6 (A.U.) Please combine Figure 4 D and E. Please change the title of Y axis; arbitrary unit (A.U.) --> NF-kB (A.U.). Please indicate "Supplementary material S2".
Author Response
Please replace the western blot image in Fig 5C, with the image from an "uncut" membrane.
We have replaced the blot image in Fig. 5C
Please change the title of Y axis in Figure 5C. arbitrary unit (A.U) --> pro-IL1b (A. U.). Figure 5A, TNFa (%) --> TNFa (Fold) or TNF-a (A.U.) Figure 5B, IL-6 (%) --> IL-6 (Fold) or IL-6 (A.U.) Please combine Figure 4 D and E. Please change the title of Y axis; arbitrary unit (A.U.) --> NF-kB (A.U.). Please indicate "Supplementary material S2".
We have made all the above requested changes